# Rethinking Patch Dependence for Masked Autoencoders

## Abstract

In this work, we examine the impact of inter-patch dependencies in the decoder of masked autoencoders (MAE) on representation learning. We decompose the decoding mechanism for masked reconstruction into self-attention between mask tokens and cross-attention between masked and visible tokens. Our findings reveal that MAE reconstructs coherent images from visible patches not through interactions between patches in the decoder but by learning a global representation within the encoder. This discovery leads us to propose a simple visual pretraining framework: cross-attention masked autoencoders (CrossMAE). This framework employs only cross-attention in the decoder to independently read out reconstructions for a small subset of masked patches from encoder outputs, yet it achieves comparable or superior performance to traditional MAE across models ranging from ViT-S to ViT-H. By its design, CrossMAE challenges the necessity of interaction between mask tokens for effective masked pretraining. Code is available here.

## 1 Introduction

Masked image modeling [46, 30, 61, 4] has emerged as a pivotal unsupervised learning technique in computer vision. One such recent work following this paradigm is masked autoencoders (MAE): given only a small, random subset of visible image patches, the model is tasked to reconstruct the missing pixels. By operating mostly on this small subset of visible tokens, MAE can efficiently pre-train high-capacity models on large-scale vision datasets, demonstrating impressive results on a wide array of downstream tasks [33, 38, 49].

The MAE framework employs *self-attention* across the entire model for self-supervised reconstruction tasks. In this setup, both masked and visible tokens engage in self-attention, not just with each other but also with themselves, aiming to generate a holistic and context-aware representation. However, the masked tokens inherently lack information. Intuitively, facilitating information exchange among adjacent masked tokens should enable the model to synthesize a more coherent image, thereby accomplishing the task of masked reconstruction and improving representation learning. A question arises, though: Is this truly the case?

We decompose the decoding process of each mask token into two parallel components: self-attention with other mask tokens, as well as cross-attention to the encoded visible tokens. If MAE relies on the self-attention with other mask tokens, its average should be on par with the cross-attention. Yet, the quantitative comparison in Figure 1.(b) shows the magnitude of mask token-to-visible token cross-attention (1.42) in the MAE decoder evaluated over the entire ImageNet validation set far exceeds that of mask token-to-mask token self-attention (0.39).

This initial observation prompts two questions: **1)** Is the self-attention mechanism among mask tokens in the decoder necessary for effective representation learning? **2)** If not, can each patch be

Submitted to 38th Conference on Neural Information Processing Systems (NeurIPS 2024). Do not distribute.

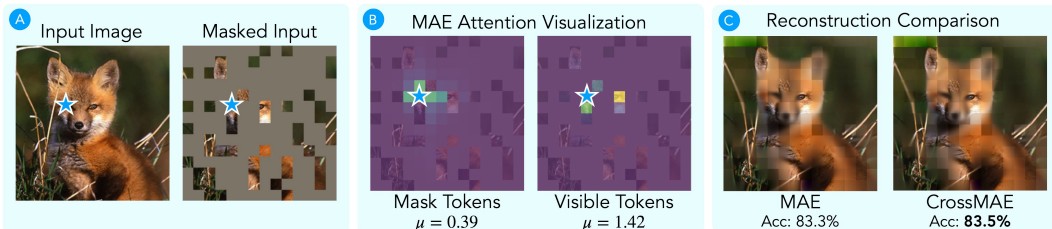

**Figure 1:** *Method Overview*. **(A)** Masked autoencoder (MAE) starts by masking random patches of the input image. **(B)** To reconstruct a mask token (marked by the blue star), MAE attends to both the masked tokens (B.Left) and the visible tokens (B.Right). A quantitative comparison over the ImageNet validation set shows that the masked tokens in MAE disproportionally attend to the visible tokens (1.42 vs 0.39), questioning the necessity of attention within mask tokens. **(C)** We propose CrossMAE, the masked patches are reconstructed from only the cross attention between the masked tokens and the visible tokens. Surprisingly, CrossMAE attains the same or better performance than MAE on ImageNet classification and COCO instance segmentation.

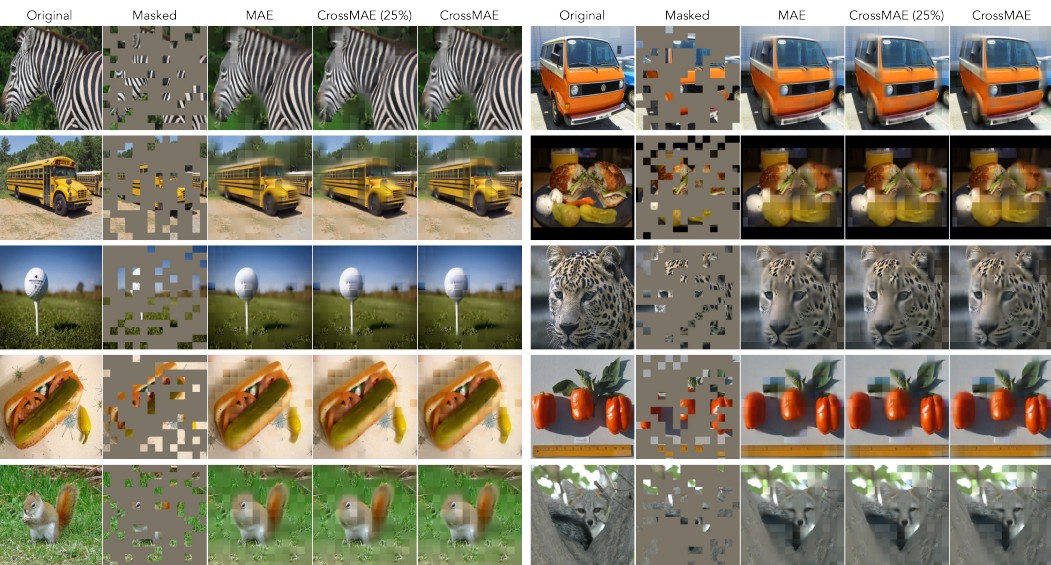

**Figure 2:** Example reconstructions of ImageNet *validation* images. For each set of 5 images, from left to right, are the original image, masked image with a mask ratio of 75%, MAE [30], CrossMAE (trained to reconstruct 25% of image tokens, or 1/3 of the mask tokens), and CrossMAE (trained to reconstruct all masked tokens). Since CrossMAE does not reconstruct them, all model outputs have the visible patches overlaid. Intriguingly, CrossMAE, when trained for partial reconstruction, can decode all mask tokens in one forward pass (shown above), indicating that the encoder rather than the decoder effectively captures global image information in its output tokens. Its comparable reconstruction quality to full-image-trained models suggests that full-image reconstruction might not be essential for effective representation learning.

*independently* read out from the encoder output, allowing the reconstruction of only a small subset of masked patches, which in turn, accelerates the pretraining without performance degradation?

In addressing these questions, we introduce CrossMAE, which diverges from MAE in three ways:

1. **Cross-attention for decoding.** Rather than passing a concatenation of mask and visible tokens to a *self-attention* decoder, CrossMAE uses mask tokens as queries to read out the masked reconstructions from the visible tokens in a *cross-attention decoder*. In this setting, mask tokens incorporate information from the visible tokens but do not interact with other mask tokens, thereby reducing the sequence length for the decoder and cutting down computational costs.

2. **Independent partial reconstruction.** With self-attention removed, the decoding of each mask token, based on the encoded features from visible tokens, becomes conditionally independent. This enables the decoding of only a fraction of masked tokens rather than the entire image.

3. **Inter-block attention.** Due to the separation of visible and mask tokens, we can use features from different encoder blocks for each decoder block. Empirically, we find solely relying on the last

encoder feature map for reconstruction, the design present in MAE, hurts feature learning. We propose a lightweight inter-block attention mechanism that allows the CrossMAE decoder to leverage a mix of low-level and high-level feature maps from the encoder, improving the learned representation.

The analysis performed on CrossMAE led to a novel way to understand MAE. Even though the patches to be reconstructed are independently decoded, our findings demonstrate that *coherent* reconstruction for each masked patch can be independently read out from the encoder output, without any interactions among masked tokens in the decoder for consistency (Figure 2). Furthermore, the downstream performance of the model remains robust even without these interactions (Figure 1.(c), Tables 1 and 2). Both pieces of evidence confirm that the encoder's output features already encapsulate the necessary global context for image reconstruction, while the decoder simply performs a readout from the encoder output to reconstruct the pixels at the location of each patch.

**To sum up, our main contributions are the following:**

1. **We present a novel understanding of MAE.** Our findings show that MAE reconstructs coherent images from visible patches *not through interactions between patches to be reconstructed* in the decoder but by *learning a global representation within the encoder*. This is evidenced by the model's ability to generate coherent images and maintain robust downstream performance without such interactions, indicating the encoder effectively captures global image information.

2. **We advocate replacing self-attention layers with a simple cross-attention readout function.** Given our discovery that the encoder in MAE already captures a comprehensive global representation, we propose replacing self-attention layers in the decoder with a more efficient information readout function. Specifically, we suggest utilizing *cross-attention* to aggregate the output tokens of the encoder into each input token within the decoder layers *independently*, thereby eliminating the need for token-to-token communication within the decoder.

3. **CrossMAE achieves comparable or superior performance with reduced computational costs** in image classification and instance segmentation compared to MAE on vision transformer models *ranging from ViT-S to ViT-H*. Code is available here.

## 2 Related Works

### 2.1 Self-Supervised Learning

In self-supervised representation learning, a model trains on a pretext task where the supervision comes from the input data itself without labels. Contrastive learning methods learn representations by contrasting positive and negative samples, such as SimCLR [11], CPC [44], MoCo [29, 12, 13], CLD [59] and SwAV [7]. Additionally, in BYOL [26], iBOT [65], DINO [8], DINOv2 [45], and MaskAlign [62] make a student model to imitate a teacher model without negative pairs.

Generative modeling, focusing on acquiring a generative model capable of capturing the underlying data distribution, is an alternative method for self-supervised learning. VAE/GAN [35] merges the strengths of variational autoencoders and generative adversarial networks to acquire disentangled representations of data. PixelCNN, PixelVAE, and PixelTransformer [55, 27, 54] generate images pixel by pixel, taking into account the context of previously generated pixels. Masked modeling, a large subclass of generative modeling, is discussed in the following subsection. After the pre-training stage, these generative models can be finetuned for many downstream applications.

### 2.2 Masked Modeling

Masked modeling learns representations by reconstructing a masked portion of the input. Pioneering works in natural language processing (NLP) present various such pretraining objectives. BERT [19] and its extensions [41, 34] use a bidirectional transformer and present few-shot learning capabilities from masked language modeling. GPT [47, 48, 5], uses autoregressive, causal masking and demonstrates multi-task, few-shot, and in-context learning capabilities.

Early works in computer vision, such as Stacked Denoising Autoencoders [57] and Context Encoder [46], investigated masked image modeling as a form of denoising or representation learning. Recently, with the widespread use of transformer [20] as a backbone vision architecture, where images are patchified and tokenized as sequences, researchers are interested in how to transfer the success in language sequence modeling to scale vision transformers. BEiT [3], MAE [30], and Sim-

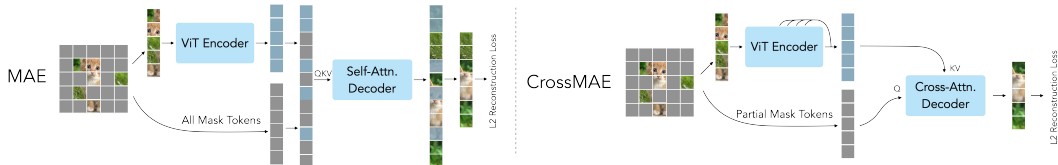

**Figure 3:** MAE [30] concatenates *all* mask tokens with the visible patch features from a ViT encoder and passes them to a decoder with self-attention blocks to reconstruct the original image. Patches that correspond to visible tokens are then dropped, and an L2 loss is applied to the rest of the reconstruction as the pretraining objective. CrossMAE instead uses cross-attention blocks in the decoder to reconstruct only a subset of the masked tokens.

MIM [61] are a few of the early works that explored BERT-style pretraining of vision transformers. Compared to works in NLP, both MAE and SimMIM [30, 61] find that a much higher mask ratio compared to works in NLP is necessary to learn good visual representation. Many recent works further extend masked pretraining to hierarchical architectures [61, 40] and study data the role of data augmentation [9, 21]. Many subsequent works present similar successes of masked pretraining for video [52, 58, 22, 28], language-vision and multi-modal pretraining [1, 39, 23] and for learning both good representations and reconstruction capabilities [60, 37].

However, BERT-style pretraining requires heavy use of self-attention, which makes computational complexity scale as a polynomial of sequence length. PixelTransformer [54] and DiffMAE [60] both use cross-attention for masked image generation and representation learning. Siamese MAE [28] uses an asymmetric masking pattern and decodes frames of a video condition on an earlier frame. In these settings, *all* masked patches are reconstructed. In this work, we investigate if learning good features necessitates high reconstruction quality and if the entire image needs to be reconstructed to facilitate representation learning. PCAE [36] progressively discards redundant mask tokens through its network, leading to a few tokens for reconstruction. VideoMAEv2 [58] concatenates randomly sampled masked tokens with visible tokens and uses self-attention to reconstruct the masked patches. In comparison, we minimally modify MAE with a cross-attention-only decoder and masked tokens are decoded in a conditional independent way.

### 2.3 Applications of Cross-Attention

In addition to the prevalent use of self-attention in computer vision, cross-attention has shown to be a cost-effective way to perform pooling from a large set of visible tokens. Intuitively, cross-attention can be seen as a parametric form of pooling, which learnably weighs different features. Touvron et al. [53] replace mean pooling with cross-attention pooling and find improvement in ImageNet classification performance. Jaegle et al. [32] uses cross-attention to efficiently process large volumes of multi-modal data. Cross-attention is also widely used for object detection. Carion et al. [6] utilizes query tokens as placeholders for potential objects in the scene. Cheng et al. [16, 15] further extend this concept by introducing additional query tokens to specifically tackle object segmentation in addition to the query tokens for object detection. Distinct from thes prior works, we are interested the role of cross-anttention for representation learning in a self-supervised manner.

## 3 CrossMAE

We start with an overview of vanilla masked autoencoders in Section 3.1. Next, in Section 3.2, we introduce the use of cross-attention in place of self-attention in the decoder for testing the necessity of interaction between mask tokens for representation learning. In Section 3.3, we discuss how eliminating self-attention in the decoding process enables us to reconstruct only a subset of masked tokens, leading to faster pretraining. Finally, Section 3.4 presents our inter-block attention mechanism, which allows decoder blocks to leverage varied encoder features.

### 3.1 Preliminaries: Masked Autoencoders

Masked Autoencoders (MAE) [30] pretrain Vision Transformers (ViTs) [20]. Each image input is first patchified, and then a random subset of the patches is selected as the visible patches. As depicted in Figure 3, the visible patches, concatenated with a learnable class token [CLS], are subsequently

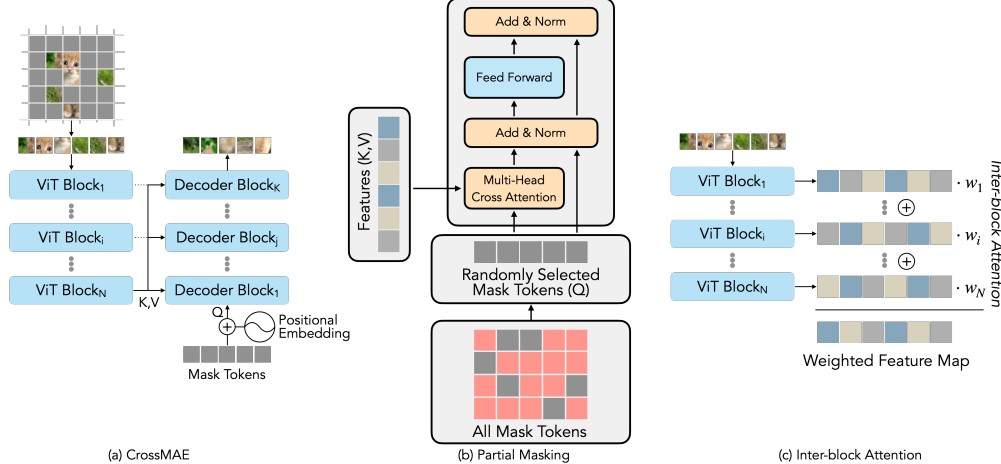

| (a) CrossMAE | (b) Partial Masking | (c) Inter-block Attention |

**Figure 4: Overview of CrossMAE. (a)** The vanilla version of CrossMAE uses the output of the last encoder block as the keys and queries for cross-attention. The first decoder block takes the sum of mask tokens and their corresponding positional embeddings as queries, and subsequent layers use the output of the previous decoder block as queries to reconstruct the masked patches. **(b)** Unlike the decoder block in [56], the cross-attention decoder block does not contain self-attention, decoupling the generation of different masked patches. **(c)** CrossMAE's decoder blocks can leverage low-level features for reconstruction via inter-block attention. It weighs the intermediate feature maps, and the weighted sum of feature maps is used as the key and value for each decoder block.

fed into the ViT encoder, which outputs a set of feature latents. The latent vectors, concatenated with the sum of the positional embeddings of the masked patches and the learnable mask token, are passed into the MAE decoder. The decoder blocks share the same architecture as the encoder blocks (i.e., both are transformer blocks with self-attention layers). Note that the number of tokens fed into the decoder is the *same* length as the original input, and the decoding process assumes that the decoded tokens depend on both visible and masked tokens. Decoder outputs pass through a fully connected layer per patch for image reconstruction. After the reconstruction is generated, the loss is applied only to the masked positions, while the reconstructions for visible spatial locations are discarded.

Recall in Sec. 1 we measure the mean attention value across all attention maps over the ImageNet validation set to study the properties of MAE. We grouped the attention values by cross-attention and self-attention between visible and masked tokens. We observed that in the decoding process of an MAE, mask tokens attend disproportionately to the class token and the visible tokens (see Figure 1.(b)). This motivates us to make design decisions and conduct experiments specifically to answer the following question: *Can we simplify the decoding process by eliminating self-attention among masked tokens without compromising the model's ability to generate coherent images and perform well on downstream tasks?*

## 3.2   Reconstruction with Cross-Attention

To address this question, we substitute the self-attention mechanism in the decoder blocks with cross-attention, using it as a readout function to decode the latent embedding from the encoder to raw pixel values. Specifically, the decoder employs multi-head cross-attention where the queries are the output from previous decoder blocks (or the sum of position embedding of the masked patches and mask token for the first decoder block). The keys and values are from the encoded features.

In the most basic CrossMAE, the output from the final encoder block is used as the key and value tokens for all layers of the decoder, as illustrated in Fig. 4(a). Further exploration in Sec.3.4 reveals that utilizing a weighted mean of selected encoder feature maps can be beneficial. The residual connections in each decoder block enable iterative refinement of decoded tokens as they progress through decoder blocks.

Diverging from the original transformer architecture [56], our decoder omits the causal self-attention layer before the introduction of multi-head cross-attention. This elimination, coupled with the fact that layer normalization and residual connections are only applied along the feature axis but not

170 the token axis, enables the independent decoding of tokens. This design choice is evaluated in the
171 ablation study section to determine its impact on performance.

172 Given the disparity in the dimensions of the encoder and decoder, MAE adapts the visible features to
173 the decoder's latent space using an MLP. However, in CrossMAE, as encoder features are integrated
174 at various decoder blocks, we embed the projection within the multi-head cross-attention module.

175 Cross-attention layers serve as a readout function that decodes the global representation provided
176 in the encoder's output tokens to the pixel values within each patch to be reconstructed. However,
177 CrossMAE does not restrict the architecture to a single cross-attention block. Instead, we stack
178 multiple cross-attention decoder blocks in a manner more akin to the traditional transformer [56].

### 3.3 Partial Reconstruction

180 The fact that CrossMAE uses cross-attention rather than self-attention in the decoder blocks brings
181 an additional benefit over the original MAE architecture. Recall that mask tokens are decoded inde-
182 pendently and thus there is no exchange of information between them, to obtain the reconstructions
183 at a specific spatial location, CrossMAE only needs to pass the corresponding mask tokens to the
184 cross-attention decoder. This allows partial reconstruction in contrast to the original full-image
185 reconstruction in the MAE architecture which needs to pass all the masked tokens as the input of the
186 decoder blocks due to the existence of self-attention in the decoder blocks.

187 To address the second question in Sec. 3.1, rather than decoding the reconstruction for all masked
188 locations, we only compute the reconstruction on a random subset of the locations and apply the loss
189 to the decoded locations. Specifically, we name the ratio of predicted tokens to all image tokens as
190 *prediction ratio* ($\gamma$), and the mask ratio ($p$). Then the prediction ratio is bounded between $\gamma \in (0, p]$.
191 Because we are sampling within the masked tokens uniformly at random and the reconstruction
192 loss is a mean square error on the reconstructed patches, the expected loss is the same as in MAE,
193 while the variance is ($p/\gamma$) times larger than the variance in MAE. Empirically, we find that scaling
194 the learning rate of MAE ($\beta$) to match the variance (i.e. setting the learning rate as $\gamma\beta/p$)) helps
195 with model performance. Since cross-attention has linear complexity with respect to the number of
196 masked tokens, this partial reconstruction paradigm decreases computation complexity. Empirically,
197 we find that the quality of the learned representations is not compromised by this approach.

### 3.4 Inter-block Attention

199 MAE combines the feature of the last encoder block with mask tokens as the input to the self-attention
200 decoder, which creates an information bottleneck by making early encoder features inaccessible
201 for the decoder. In contrast, CrossMAE's cross-attention decoder decouples queries from keys and
202 values. This decoupling allows different cross-attention decoder blocks to take in feature maps from
203 different encoder blocks. This added degree of flexibility comes with a design choice for selecting
204 encoder features for each decoder block. One naive choice is to give the feature of the $i$th encoder
205 block to the last $i$th decoder (*e.g.*, feeding the feature of the first encoder to the last decoder), in a
206 U-Net-like fashion. However, this assumes the decoder's depth matches the depth of the encoder,
207 which is not the case for MAE or CrossMAE.

208 Instead of manually matching each decoder block with an encoder feature map, we make the selection
209 *learnable* and propose inter-block attention for feature fusion for each decoder block (Figure 4(c)).
210 Analogous to the inter-patch cross-attention that takes a weighted sum of the visible token embeddings
211 across the patch dimensions to update the embeddings of masked tokens, inter-block attention takes
212 a weighted sum of the visible token embeddings *across different input blocks* at the same spatial
213 location to fuse the input features from multiple blocks into one feature map for each decoder block.

214 Concretely, each decoder block takes a weighted linear combination of encoder feature maps $\{f_i\}$ as
215 keys and values. Specifically, for each key/value token $t_k$ in decoder block $k$ in a model with encoder
216 depth $n$, we initialize a weight $w^k \in \mathcal{R}^n \sim \mathcal{N}(0, 1/n)$. Then $t_k$ is defined as

$$t_k = \sum_{j=1}^{n} w_j^k f_j. \tag{1}$$

217
218 In addition to feature maps from different encoder blocks, we also include the inputs to the first
219 encoder block to allow the decoder to leverage more low-level information to reconstruct the original

| Method | ViT-S | ViT-B | ViT-L | ViT-H |
|---|---|---|---|---|
| Supervised [50] | 79.0 | 82.3 | 82.6 | 83.1 |
| DINO [8] | - | 82.8 | - | - |
| MoCo v3 [14] | 81.4 | 83.2 | 84.1 | - |
| BEiT [3] | - | 83.2 | 85.2 | - |
| MultiMAE [2] | - | 83.3 | - | - |
| MixedAE [9] | - | 83.5 | - | - |
| CIM [21] | **81.6** | 83.3 | - | - |
| MAE [30] | 78.9 | 83.3 | **85.4** | 85.8 |
| CrossMAE (25%) | 79.2 | 83.5 | 85.4 | **86.3** |
| CrossMAE (75%) | 79.3 | **83.7** | 85.4 | - |

**Table 1:** *ImageNet-1K classification accuracy.* CrossMAE performs on par or better than MAE. All experiments are run with 800 epochs. The best results are in **bold** while the second best results are underlined.

| Method | $AP^{box}$ | | $AP^{mask}$ | |
|---|---|---|---|---|
| | ViT-B | ViT-L | ViT-B | ViT-L |
| Supervised [38] | 47.6 | 49.6 | 42.4 | 43.8 |
| MoCo v3 [14] | 47.9 | 49.3 | 42.7 | 44.0 |
| BEiT [3] | 49.8 | 53.3 | 44.4 | 47.1 |
| MixedAE [9] | 50.3 | - | 43.5 | - |
| MAE [38] | 51.2 | 54.6 | 45.5 | 48.6 |
| CrossMAE | **52.1** | **54.9** | **46.3** | **48.8** |

**Table 2:** *COCO instance segmentation.* Compared to previous masked visual pretraining works, CrossMAE performs favorably on object detection and instance segmentation tasks.

image. We can select a subset of the feature maps from the encoder layers instead of all feature maps. This reduces the computation complexity of the system. We ablate this in Table 3d.

We show that using the weighted features rather than simply using the features from the last block greatly improves the performance of CrossMAE. Intriguingly, in the process of learning to achieve better reconstructions, early decoder blocks tend to prioritize information from later encoder blocks, while later decoder blocks focus on earlier encoder block information, as demonstrated in Section 4.5.

# 4 Experiments

We perform self-supervised pretraining on ImageNet-1K, following MAE [30]'s hyperparameter settings, only modifying the learning rate and decoder depth. The hyperparameters were initially determined on ViT-Base and then directly applied to ViT-Small, ViT-Large, and ViT-Huge. Both CrossMAE and MAE are trained for 800 epochs. We provide implementation details and more experiments in the appendix.

## 4.1 ImageNet Classification

**Setup.** The model performance is evaluated with end-to-end fine-tuning, with top-1 accuracy used for comparison. Same as in Figure. 2, we compare two versions of CrossMAE: one with a prediction ratio of 25% (1/3 of the mask tokens) and another with 75% (all mask tokens). Both models are trained with a mask ratio of 75% and a decoder depth of 12.

**Results.** As shown in Table 1, CrossMAE outperforms vanilla MAE using the same ViT-B encoder in terms of fine-tuning accuracy. This shows that replacing the self-attention with cross-attention *does not degrade* the downstream classification performance of the pre-trained model. Moreover, CrossMAE outperforms other self-supervised and masked image modeling baselines, *e.g.*, DINO [8], MoCo v3 [14], BEiT [3], and MultiMAE [2].

## 4.2 Object Detection and Instance Segmentation

**Setup.** We additionally evaluate models pretrained with CrossMAE for object detection and instance segmentation, which require deeper spatial understanding than ImageNet classification. Specifically, we follow ViTDet [38], a method that leverages a Vision Transformer backbone for object detection and instance segmentation. We report box AP for object detection and mask AP for instance segmentation, following MAE [30]. We compare against supervised pre-training, MoCo-v3 [14], BEiT [4], and MAE [30].

**Results.** As listed in Table 2, CrossMAE, with the default 75% prediction ratio, performs better compared to these baselines, including vanilla MAE. This suggests that similar to MAE, CrossMAE performance on ImageNet positively correlates with instance segmentation. Additionally, Cross-MAE's downstream performance scales similarly to MAE as the model capacity increases from ViT-B to ViT-L. This observation also supports our hypothesis that partial reconstruction is suprisingly sufficient for learning dense visual representation.

| Method | Acc. (%) | | Mask Ratio | Acc. (%) | | Pred. Ratio | Acc. (%) |
|---|---|---|---|---|---|---|---|
| MAE | 83.0 | | 65% | **83.5** | | 15% | 83.1 |
| CrossMAE | **83.3** | | 75% | 83.3 | | 25% | 83.2 |
| CrossMAE + Self-Attn | 83.3 | | 85% | 83.3 | | 75% | **83.3** |

**(a)** **Attention type** in decoder blocks. Adding back self-attention between mask tokens does not improve performance.

**(b)** **Mask ratio.** CrossMAE has consistent performance across high mask ratios.

**(c)** **Prediction ratio.** CrossMAE performs well even when only a fraction of mask tokens are reconstructed.

| # Feature Maps Fused | Acc. (%) | | Decoder Depth | Acc. (%) | | Image Resolution | Acc. (%) |
|---|---|---|---|---|---|---|---|
| 1 | 82.9 | | 1 | 83.0 | | 224 | 83.2 |
| 3 | 83.3 | | 4 | 83.1 | | 448 | **84.6** |
| 6 | **83.5** | | 8 | 83.1 | | | |
| 12 | 83.3 | | 12 | **83.3** | | | |

**(d)** **Inter-block attention.** A combination of six select encoder feature maps is best.

**(e)** **Decoder depth.** CrossMAE performance scales with decoder depth.

**(f)** **Input resolution.** CrossMAE scales to longer input sequences.

Table 3: *Ablations on CrossMAE*. We report fine-tuning performance on ImageNet-1K classification with 400 epochs (*i.e.*, half of the full experiments) with ViT-B/16. MAE performance is reproduced using the official MAE code. Underline indicates the default setting for CrossMAE. **Bold** indicates the best hyperparameter among the tested ones. 1 feature map fused (row 1, Table 3(d)) indicates using only the feature from the last encoder block. We use 25% prediction ratio for both settings in Table 3(f) to accelerate training.

## 4.3 Ablations

**Cross-Attention vs Self-Attention.** As shown in Table 3a, CrossMAE, with its cross-attention-only decoder, outperforms vanilla MAE in downstream tasks as noted in Section 4.1. Additionally, combining cross-attention with self-attention does not enhance fine-tuning performance, indicating that cross-attention alone is adequate for effective representation learning.

**Mask Ratio and Prediction Ratio.** In our experiments with different mask and prediction ratios (*i.e.*, the ratio of mask tokens to all tokens and the ratio of reconstructed tokens to all tokens, respectively) (see Table 3b and Table 3c), we found that our method's performance is not significantly affected by variations in the number of masked tokens. Notably, CrossMAE effectively learns representations by reconstructing as few as 15% of tokens, compared to the 100% required by vanilla MAE, with minimal impact on downstream fine-tuning performance, which shows that partial reconstruction is sufficient for effective representation learning.

**Inter-block Attention.** Our ablation study, detailed in Table 3d, explored the impact of varying the number of encoder feature maps in our inter-block attention mechanism. We found that using only the last feature map slightly lowers performance compared to using all 12. However, even a partial selection of feature maps improves CrossMAE's performance, with the best results obtained using 6 feature maps. This indicates that CrossMAE does not require all features for optimal performance.

**Decoder Depth.** Table 3e shows that a 12-block decoder slightly improves performance compared to shallower ones. Remarkably, CrossMAE achieves similar results to MAE with just one decoder block, demonstrating its efficiency. Our experiments in Figure 7 that models with lower prediction ratios benefit more from deeper decoders.

**Input Resolution.** We extend CrossMAE to longer token lengths by increasing the image resolution with constant patch size. Escalating the resolution from 224 to 448 increases the token length from 197 to 785, challenging the scalability of current approaches. Thus, we opt for a CrossMAE variant with a 25% prediction ratio. In Table 3f, we observe that the classification accuracy positively correlates with the input resolution, indicating that CrossMAE can scale to long input sequences.

## 4.4 Training Throughput and Memory Utilization

Due to partial reconstruction and confining attention to between mask tokens and visible tokens, CrossMAE improves pre-training efficiency over MAE. Results in Table 10 show that the FLOPs

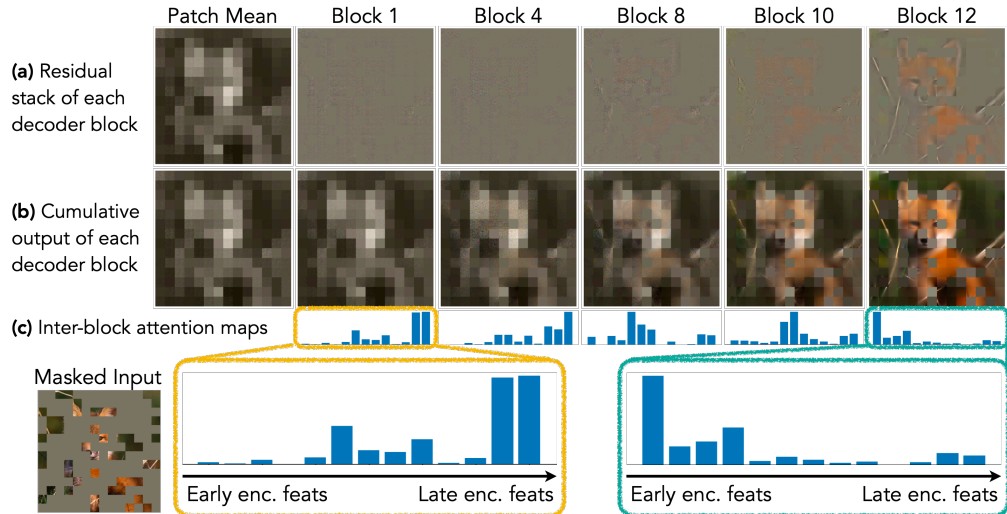

**Figure 5:** We visualize the output of each decoder block. (a-b) **Different decoder blocks play different roles in the reconstruction**, with most details emerging at later decoder blocks, which confirms the motivation for inter-block attention. (c) Visualizations of inter-block attention shows that **different decoder blocks indeed attend to feature from different encoder blocks**, with later blocks focusing on earlier encoder features to achieve reconstruction. The reconstructions are unnormalized w.r.t ground truth mean and std for each patch.

reduction does translate to an $1.54\times$ training throughput and at least 50% reduction in GPU memory utilization compared to MAE.

## 4.5 Visualizations

**Visualizing Per-block Reconstruction.** Rather than only visualizing the final reconstruction, we have two key observations that allow us to visualize the work performed by each decoder block: **1)** Transformer blocks have skip connections from their inputs to outputs. **2)** The final decoder block's output goes through a linear reconstruction head to produce the reconstruction. As detailed in Appendix D, we can factor out each block's contribution in the final reconstruction with linearity.

This decomposition allows expressing the reconstruction as an image stack, where summing up all the levels gives us the final reconstruction. As shown in Figure 5 (a,b), we observe that different decoder blocks play different roles in reconstruction, with most details emerging at later decoder blocks. This justifies the need for low-level features from early encoder blocks, motivating inter-block attention.

**Visualizing Inter-block Attention Maps.** As shown in the visualizations of the attention maps of inter-block attention in 5(c), CrossMAE naturally leverages the inter-block attention to allow the later decoder blocks to focus on earlier encoder features to achieve reconstruction and allow the earlier decoder blocks to focus on later encoder features. This underscores the necessity for different decoder blocks to attend to different encoder features, correlating with the performance improvements when inter-block attention is used.

## 5 Discussion and Conclusion

In our study, we present a novel understanding of MAE, demonstrating that coherent image reconstruction is achieved not through interactions between patches in the decoder but by learning a global representation within the encoder. Based on this insight, we propose replacing self-attention layers in the decoder with a simple readout function, specifically utilizing cross-attention to aggregate encoder outputs into each input token within the decoder layers independently. This approach, tested across models ranging from ViT-S to ViT-H, achieves comparable or better performance in image classification and instance segmentation with reduced computational requirements, showcasing the potential for more efficient and scalable visual pretraining methods. Our findings underscore the efficacy of the encoder's global representation learning, paving the way for streamlined decoder architectures in future MAE implementations. CrossMAE's efficiency and scalability demonstrate potential for large-scale visual pretraining, particularly on underutilized in-the-wild video datasets. However, our work has not yet explored scaling to models larger than ViT-H, the largest model examined in MAE, leaving this for future research.

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

# A   Implementation details

## A.1   Attention Calculation

To compare the attention values for mask tokens in vanilla MAE (Figure 1), we trained a ViT-B/16 MAE for 800 epochs using the default hyperparameters provided in [30]. For each image, we randomly generate a 75% binary mask ($m$) for all tokens, with $m_i = 1$ representing a token being masked and $m_i = 0$ otherwise. During the forward pass of the decoder, for each self-attention operation, the attention map is stored. This means that for the default MAE, a total of 8 attention maps, each with 16 attention heads are stored. Based on the mask pattern, we calculate the outer product ($m \cdot m^\top$) for the self-attention among mask tokens, and $m \cdot (1 - m^\top)$ for the cross-attention from the mask token to the visible tokens. We then calculate the average across all feature maps and attention heads for self-attention and cross-attention to get the image average values. Lastly, we averaged across the entire ImageNet validation set to obtain the final values.

## A.2   Inter-Block Attention

We tried a few implementations for inter-block attention (IBA) and found the following implementation to be the fastest and most memory-efficient. In this implementation, we combine inter-block attention for all encoder layers as a single forward pass of a linear layer. For each decoder block, we index into the output tensor to extract the corresponding feature map, and a layer norm will be applied before the feature map is fed into the decoder block. Other alternatives we tried include 1) performing separate inter-block attentions before each decoder block, and 2) 1x1 convolution on the stacked encoder feature maps.

In MAE, there exists a layer norm after the last encoder feature map before feeding into the decoder. In our implementation, we only add layer norm after inter-block attention. We find that adding an additional layer norm before inter-block attention to each encoder feature map does not lead to improvements in model performance but will significantly increase GPU memory usage.

The pseudo-code of inter-block attention is the following:

```python
class InterBlockAttention():
    def __init__(self, num_feat_maps, decoder_depth):
        self.linear = Linear(num_feat_maps, decoder_depth, bias=False)
        std_dev = 1. / sqrt(num_feat_maps)
        init.normal_(self.linear.weight, mean=0., std=std_dev)

    def forward(self, feature_maps : list):
        """
        feature_maps: a list of length num_feat_maps, each with
    dimension
        Batch Size x Num. Tokens x Embedding Dim.
        """
        stacked_feature_maps = stack(feature_maps, dim=-1)
        return self.linear(stacked_feature_maps)
```

Additionally, we further investigate the importance of using a cross-attention decoder, where each decoder block can use different feature maps from the encoder for decoding. In this experiment, we incorporated IBA into MAE, which uses only a self-attention decoder. Specifically, we concatenate the interblock attention features with the masked tokens. We then feed the combined features into MAE's self-attention decoder. We pre-trained the model and finetuned it for Imagenet classification. The results are presented in Table. 4, where all models are pre-trained for 400 epochs. We observe that inter-block attention has negligible performance improvements for MAE, potentially because MAE only takes in one feature map in its decoder. In contrast, inter-block attention allows cross-attention layers in CrossMAE to attend to features from different encoder blocks, thanks to its decoupling of queries with keys and values.

## A.3   Ablation that Adds Self-Attention

In Section 4.3 (a), we propose adding self-attention back to CrossMAE as an ablation. In that particular ablation study, we analyze the effect of self-attention between the masked tokens, which

| Method | Acc. (%) |
|---|---|
| MAE | 83.0 |
| MAE + IBA | 83.0 |
| CrossMAE (25%) | 83.2 |
| CrossMAE (75%) | **83.3** |

**Table 4:** For MAE, inter-block attention has very small differences in terms of finetuning performance, potentially due to the fact that MAE's decoder only takes in one set of features.

can be used to improve the consistency for reconstruction. Specifically, we modify the formulation in the original transformer paper [56], where the mask/query tokens are first passed through a multi-head self-attention and a residual connection before being used in the multiheaded cross-attention with the features from the encoder. The primary difference with the vanilla transformer decoder implementation [56] is we do not perform casual masking in the multi-head self-attention. Please reference Figure 6 for a more visual presentation of the method.

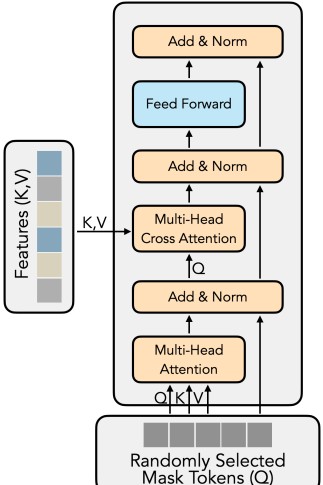

**Figure 6:** Modification for self-attention ablation

## A.4 Ablation on Inter-block Attention

In Table 3d, the following cases are considered. 1 feature map (row 1) does not use inter-block attention. Each decoder block only takes the last feature map from the encoder as the keys and values. For scenarios where more than one feature map is used, the output of the patch embedding (input to the ViT) is also used.

In addition to the simple design of inter-block attention proposed above, we also experimented with a variant of inter-block attention by further parameterizing the attention with linear projections. Specifically, rather than directly performing weighted sum aggregation to form the features for each cross-attention layer in the decoder, we added a linear projection for each encoder feature before the feature aggregation. We denote this variant as *CrossMAE+LP*. As shown in the Table. 5 (with ViT-B pre-trained for 400 epochs, consistent with the setting in Table. 3), adding a linear projection slightly improves the performance. This indicates that it is possible to design variants of readout functions, such as through improved inter-block attention, to improve the feature quality of CrossMAE.

| Method | Acc. (%) |
|---|---|
| CrossMAE | 83.3 |
| CrossMAE + LP | **83.5** |

**Table 5:** Improving inter-block attention by adding linear projections to the input features. The performance gain indicates that it is possible to design variants of readout functions to improve CrossMAE.

## A.5 Hyperparameters

**Pre-training**: The default setting is in Table 6, which is consistent with the official MAE [30] implementation. As mentioned in Sec. 3.4, we scale the learning rate by the ratio between mask ratio ($p$) and prediction ratio ($\gamma$) to ensure the variance of the loss is consistent with [30]. Additionally, we use the linear learning rate scaling rule [25]. This results in $lr = \gamma * base\_lr * batchsize/(256 * p)$. For Table 1, we use 12 decoder blocks, with mask ratio and prediction ratio both 75%, and interblock attention takes in all encoder feature maps. For the 400 epochs experiments in Table 2, we scale the warm-up epochs correspondingly. Other hyperparameters, such as decoder block width, are the same as MAE.

**Finetuning**: We use the same hyperparameters as MAE finetuning. We use global average pooling for finetuning. In MAE, the layer norm for the last encoder feature map is removed for finetuning, which is consistent with our pretraining setup. Please refer to Table 7 for more detail.

| Config | Value |
|---|---|
| optimizer | AdamW [43] |
| base learning rate | 1.5e-4 |
| learning rate schedule | cosine decay [42] |
| batch size | 4096 |
| weight decay | 0.05 |
| optimizer momentum | $\beta_1, \beta_2 = 0.9, 0.95$ [10] |
| warm up epoch [24] | 20, 40 |
| total epochs | 400, 800 |
| augmentation | RandomResizedCrop, RandomHorizontalFlip |

**Table 6:** Pretraining Hyperparameters

## A.6 Compute Infrastructure

Each of the pretraining and finetuning experiments is run on 2 or 4 NVIDIA A100 80GB GPUs. The batch size per GPU is scaled accordingly and we use gradient accumulation to avoid out-of-memory errors. ViTDet [38] experiments use a single machine equipped with 8 NVIDIA A100 (80GB) GPUs. We copy the datasets to the shared memory on the machines to accelerate dataloading. We use FlashAttention-2 [18] to accelerate attention calculation.

| Config | Value |
|---|---|
| optimizer | AdamW |
| base learning rate | 1e-3 |
| learning rate schedule | cosine decay |
| batch size | 1024 |
| weight decay | 0.05 |
| optimizer momentum | $\beta_1, \beta_2 = 0.9, 0.999$ |
| warm up epoch | 5 |
| total epochs | 100 (B), 50 (L) |
| augmentation | RandAug (9, 0.5) [17] |
| label smoothing [51] | 0.1 |
| mixup [64] | 0.8 |
| cutmix [63] | 1.0 |
| drop path [31] | 0.1 |

**Table 7:** Finetuning Hyperparameters

## B  Additional Experiments

### B.1  Linear Probe

We provide linear probe comparisons (at 800 epochs) for ViT-Small and ViT-Base in Table. 8. For both of these experiments, we run CrossMAE with a prediction ratio of 75% (reconstruction of all masked patches). These results show that CrossMAE achieves slightly better linear probe performance than vanilla MAE.

| Method | ViT-S | ViT-B |
|---|---|---|
| MAE | 49.7 | 65.1 |
| CrossMAE | **51.5** | **65.4** |

**Table 8:** Linear probe experiments of CrossMAE.

### B.2  Masking Strategy

| Method | Acc. (%) |
|---|---|
| Grid Masking | 83.2 |
| Random Masking | **83.3** |

**Table 9:** Ablation of masking strategies.

Similar to MAE [30], we here ablate the masking pattern. Instead of random masking, we perform grid-wise sampling that "keeps one of every four patches" (see MAE Figure 6). The finetuning performance is reported in Table. 9 for ViT-B (at 400 epochs), which shows that grid masking does not lead to additional improvements in downstream performance.

## C  Runtime and GPU Memory Comparisons with MAE

| Method | Memory (MB/GPU) | Runtime (min/epoch) | Acc. (%) |
|---|---|---|---|
| MAE | OOM (>81920) | 5.19* | 83.3 |
| CrossMAE | **41177** | **3.38** | **83.5** |

**Table 10: CrossMAE greatly improves the training throughput and reduces the memory requirements**, lowering the barrier for masked pretraining. Statistics are measured on 2 NVIDIA A100 80GB GPUs. Please refer to Appendix C for comparison details. *: MAE's default batch size exceeds the capacity of 4 GPUs, requiring gradient accumulation for runtime measurement.

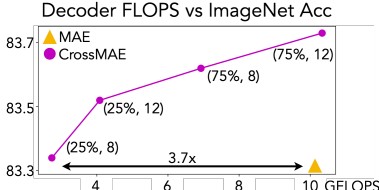

**Figure 7:** We compare ViT-B which is pre-trained for 800 epochs with different variants of Cross-MAE v.s. MAE. For CrossMAE, we vary the prediction ratio $p$ and number of decoder blocks $n$, and we denote each as $(p, n)$. While all experiments are run with inter-block attention, Cross-MAE has lower decoder FLOPS than MAE [30] and performs on par or better.

All experiments in Table 10 are conducted on a server with 4 NVIDIA A100 (80GB) GPUs, with the standard hyperparameters provided above for pretraining. NVLink is equipped across the GPUs. We use the default setting for MAE and set the global batch size to 4096. For CrossMAE, we also use the default setting with a prediction ratio 0.25, and this takes around 41GB memory per GPU without gradient accumulation (i.e., local batch size is set to 1024 samples per GPU). However, the same local batch size results in out-of-memory (OOM), which indicates that the total memory requirement is larger than the available memory for each GPU (80GB). To run MAE on same hardware, we thus employ gradient accumulation with a local batch size of 512 to maintain the global batch size. The benchmark runs each method and measures the average per epoch runtime as well as the max memory allocation for 10 training epochs. Our experiments in Figure 7 show that models with lower prediction ratios benefit more from deeper decoders. Our model performs on par or better when compared to MAE, with up to 3.7× lower decoder FLOPS.

## D Visualizing the Contributions per Decoder Block

We propose a more fine-grained visualization approach that allows us to precisely understand the effect and contribution of each decoder block.

Two key observations enable per-block visualization: **1)** Transformer blocks have residual connections from their inputs to outputs. Let $f_i$ be the output and $g_i(\cdot)$ the residual function of decoder $i$, so $f_i = f_{i-1} + g_i(f_{i-1})$. **2)** The final decoder block's output goes through a reconstruction head $h$, which is linear, consisting of a layer-norm and a linear layer, to produce the reconstruction. With $D$ as the decoder depth, $f_0$ the initial input, and $y$ the final output, $y$ is recursively defined as $y = h(f_{D-1} + g_D(f_{D-1}))$, which simplifies due to the linearity of $h$:

$$
\begin{aligned}
\mathbf{y} &= h(f_0 + g_1(f_0) + \cdots + g_D(f_{D-1})) \\
&= \underbrace{h(f_0)}_{\text{Pos Embed. + Mask Token}} + \underbrace{h(g_1(f_0))}_{\text{Block 1}} + \cdots + \underbrace{h(g_D(f_{D-1}))}_{\text{Block D}}
\end{aligned}
$$

This decomposition allows us to express the reconstruction as an image stack, where the sum of all the levels gives us the final reconstruction. We present the visualization in Figure 5.

