# OpenReview forum: "Rethinking Patch Dependence for Masked Autoencoders"
_NeurIPS.cc/2024/Conference — Submitted to NeurIPS 2024_

### Official Review · Reviewer_yYWM · 2024-07-07

**Soundness:** 2
**Presentation:** 3
**Contribution:** 2
**Rating:** 4
**Confidence:** 4

**Summary:**

This paper reveals the role of inter-patch dependencies in the decoder of MAE on representation learning. The paper shows that MAE achieves coherent image reconstruction through global representations learned in the encoder rather than interactions between patches in the decoder. Based on this, the authors propose CrossMAE, which only utilizes cross-attention in the decoder.

**Strengths:**

- The approach of analyzing the reconstruction process through self-attention between mask tokens and cross-attention between mask and visible tokens is intriguing.
- The writing is clear and easy to follow, with main messages that are solid and insightful.

**Weaknesses:**

1. Idea/Novelty
- The claim that MAE reconstruction is achieved through global representation learning within the encoder rather than interactions between patches needs more support. Recent studies linking MAE to contrastive learning have found that the receptive field of specific mask tokens in the decoder is relatively small. Could the role of mask tokens in the decoder be to capture local area information? This might explain the smaller attention magnitude of masked tokens compared to visible tokens in Figure 1(b).
- There is a concern that without self-attention (i.e., with the proposed method), the observation that authors made on the vanilla MAE may no longer be valid. Additional explanation on this point is necessary as this observation is the main motivation for suggesting CrossMAE.

2. Additional justification
- Effectiveness of using a subset of mask tokens as queries: Unlike the traditional architecture, this method uses only a subset of mask tokens as queries. Detailed analysis and interpretation are needed on why this is effective.
- Performance differences when using the entire set of mask tokens versus a subset (and what percentage of mask tokens is used) should be reported.

3. Experiment
- For a fair comparison, CrossMAE's performance should be evaluated using the same setting as the original MAE, especially regarding the fine-tuning recipe.
- The current experimental results do not convincingly demonstrate the effectiveness of the method. For classification tasks, only the linear-probing and fine-tuning results on IN1K are reported. Following the previous works, classification on various downstream datasets should be also considered.
- For generalizability, evaluation on another task like semantic segmentation (e.g. on ADE20K) would be useful to verify that the suggested method learns the generalizable feature representation.

**Questions:**

Please refer to the weakness section.

**Limitations:**

The authors have not discussed limitations of this work except for the very last sentence of section 5, indicating that they have discussed limitations in this section in the questionnaire #2. It is strongly recommended to disclose more detailed limitations of the proposed work.

---

> ### Author Rebuttal · Authors · 2024-08-07
>
> Thank you for the review and we appreciate these suggestions. We have performed the requested experiments and will revise the paper accordingly.
>
> > [W1] “The claim that MAE reconstruction is achieved through global representation learning within the encoder rather than interactions between patches needs more support.”
>
> The interpretation seems inconsistent with what was proposed in the work. We demonstrate that interactions between masked and visible patches are the primary contributors to MAE reconstruction, while interactions among masked patches contribute marginally. Figure 1(b) supports this, showing masked tokens attend much more strongly to visible tokens than to other masked tokens.
>
> We further validated this with two test-time ablations:
> 1. Disabling self-attention among masked tokens still produces valid reconstructions (Figure I for Reviewer yYWM).
> 2. Disabling cross-attention fails to reconstruct the image (Figure II for Reviewer yYWM).
>
> These results confirm that vanilla MAE primarily uses cross-attention for reconstructions, not self-attention between masked patches.
>
> > [W1] “Could the role of masked tokens in the decoder be to capture local area information? This might explain the smaller attention magnitude of masked tokens compared to visible tokens in Figure 1(b).”
>
> We apologize for the potential misunderstanding. The hypothesis that each masked token attends to tokens nearby is also supported by our empirical analysis in Figure 1. However, this observation  **does not contradict our claims** that **1)** all the encoder features of visible patches collectively construct a representation of the whole image, including the parts that were masked out, and **2)** the masked patches attend to visible patches more than they attend to other masked patches.
>
> For the first claim, we note that each masked patch can be reconstructed by attending to the encoder features of the visible patches, indicating the encoder features for the visible parts **collectively** contain information for **the whole image**. In this way, **we can independently decode each masked patch without considering the pixel values of other masked patches**. We will revise the wording to reduce confusion.
>
> For the second claim, although the masked tokens often attend to tokens nearby, they still attend to visible tokens much more strongly, justified by Figure 1(b).
>
> > [W2] “There is a concern that without self-attention (i.e., with the proposed method), the observation that authors made on the vanilla MAE may no longer be valid.”
>
> We observed that in vanilla MAE, self-attention among masked tokens is much weaker than cross-attention between masked and visible tokens. Converting the self-attention MAE to a cross-attention-only decoder changes the attention mask to allow only attention from masked tokens to visible tokens (i.e., **equivalent to setting self-attention magnitude to 0**), aligning with our previous observation.
>
> > [W3] “Detailed analysis and interpretation are needed on why (using only a subset of masked tokens as queries) is effective.”
>
> The intention and justification for using a prediction ratio that is different from the mask ratio are presented in the general response.
>
> > [W4] Performance differences when using the entire set of masked tokens versus a subset (and what percentage of masked tokens is used) should be reported.
>
> We will revise to emphasize that the downstream performance tradeoff is provided in Table 1 and ablated in Table 3(c). The FLOPS efficiency tradeoff is provided in the general response and will be added to our work.
>
> > [W5] For a fair comparison, CrossMAE's performance should be evaluated using the same setting as the original MAE, especially regarding the fine-tuning recipe.
>
> We use the same fine-tuning and evaluation recipe as outlined in MAE. Hyperparameters such as decoder block width and the number of training epochs are the same as MAE for a fair comparison.
>
> > [W6] “Following the previous works, the classification on various downstream datasets should be also considered.”
>
> We provide more experiments on iNaturalist2019 and Places365 (ViT-B, pretrained for 800 epochs). **We find that CrossMAE performs comparably to MAE for transfer learning.**
>
> | | MAE | CrossMAE (0.25) | CrossMAE (0.75) |
> |-|-|-|-|
> | iNaturalist2019 Accuracy | 79.8 | 79.4 | **80.1** |
>
> | | MAE | CrossMAE (0.25) | CrossMAE (0.75) |
> |-|-|-|-|
> | Places365 Accuracy | **57.9** | **57.9** | 57.8 |
>
> > [W7] “For generalizability, evaluation on another task like semantic segmentation (e.g. on ADE20K).”
>
> In addition to COCO instance segmentation, we provide semantic segmentation results on ADE20K, which further demonstrates that CrossMAE learns generalizable features.
>
> | | MAE | CrossMAE (0.25) | CrossMAE (0.75) |
> |-|-|-|-|
> | ADE20k mIoU | 47.7 | 47.7 | **48.1** |
>
> > [Limitations] “The authors have not discussed the limitations of this work except for the very last sentence of section 5”
>
> We will add more limitations in the revision:
>
> While CrossMAE improves the efficiency of pretraining, it still inherits the limitations of MAE. For example, although CrossMAE performs on par or better than MAE with the same hyperparameters, as ablated in Table 3, finding the optimal masking and prediction ratio can require more experimentation on new datasets. In addition, since CrossMAE still follows a masked reconstruction objective, the learned model predicts content based on the training dataset and will reflect biases in those data. Finally, both MAE and CrossMAE only work on vision transformers and their variants, while adapting them to CNN-based methods is non-trivial and may require custom CUDA kernels to maintain efficiency.
>
> **Given the clarifications we’ve provided and the promising results in many requested experiments, we kindly ask if you would consider raising your assessment score. Additionally, please let us know if there are any new concerns or further questions we can address for you!**

---

> > ### Author Response · Authors · 2024-08-12
> > **A gentle reminder - 2 days left for the author-reviewer discussion**
> >
> > Dear reviewer,
> >
> > We wanted to ask if you had a chance to check our response with additional details and clarifications, analyses on transfer learning to different datasets and task, and an updated limitations section.
> >
> > Please also consider checking our general response, which includes a more detailed analysis of the improved efficiency of CrossMAE.
> >
> > Please let us know if we addressed your concerns and/or if any further information or experiments would be helpful. We would be happy to provide them.
> >
> > Many thanks!
> > Authors

---

> > > ### Author Response · Authors · 2024-08-13
> > > **Gentle reminder - 1 day left for the author-reviewer discussion**
> > >
> > > Dear reviewer,
> > >
> > > Please let us know if our response addressed your concerns and/or if you have any additional questions. We would be happy to answer them before the end of the discussion period.
> > >
> > > Thanks!
> > >
> > > Authors

---

### Official Review · Reviewer_cyeA · 2024-07-11

**Soundness:** 3
**Presentation:** 3
**Contribution:** 2
**Rating:** 5
**Confidence:** 4

**Summary:**

The paper introduces a novel pre-training approach called CrossMAE. Instead of concatenating the masked and visible tokens for the decoder, the authors add cross-attention to decode the masked tokens by using them and the visible patch embeddings as separate inputs to the decoder. Further, the authors introduce a method to only partially reconstruct the masked patches, and leverage inter-bock attention to fuse feature across different layers.

**Strengths:**

- The paper is well motivated through a practical observation
- The authors propose a useful technical contribution which seem intuitive given the described observations
- The paper is well written and technically sound
- All visualizations provide additional value, I especially like Figure 5. It describes the effect of the contributions well
- Judging from the experiment section, the presented approach mostly improves over the vanilla MAE and other MAE-like follow-up works

**Weaknesses:**

- I feel like the paper is missing a more structure ablation of the individual contributions. I think the paper would benefit from having a simple table where all contributions are added sequentially to better identify the performance effect of the individual contributions as in:
	MAE X.X
	+ Cross-Attn X.X
	+ Partial Reconstruction X.X
	+ Inter-Block Attn X.X
- As can be observed from Table 3 c), the final setting (underlined) of the prediction ratio, 0.75, turns out to be exactly the same as the optimal masking ratio, 0.75. If I understood correctly, this means that in practice, CrossMAE works best when it predicts all tokens that were masked, not just a fraction of them. Only predicting part of the masked tokens was previously listed as a contribution. Therefore, I don’t understand how this additional hyper parameter provides any benefit for better downstream performance. Maybe I’m missing something and this be cleared up by answering the previous point.
- All models are only trained for 800 epochs. The original MAE reaches peak performance at 1600 epochs. For a thorough comparison, it would be necessary to also train CrossMAE for 1600 epochs and see if the performance gains sustain, or if performance has peaked at 800 epochs.
- Table 1 is missing the CrossMAE ViT-H with 75% masking ratio
- Contribution 2 and 3 don’t seem to be as well motivated in the introduction in comparison to Contribution 1
- Better performance is listed as a contribution. IMO this is not a contribution, rather a result of the technical contributions

**Questions:**

I like the idea and motivation of the paper. It starts from an interesting observation of the vanilla MAE, and aims to improve this aspect. Unfortunately, it is not fully clear which of the proposed contributions actually have an impact on performance. Table 3a) shows that adding Cross-Attn improves downstream performance. But since the authors choose the masking ratio to be the same as the prediction ratio, there doesn’t seem to be an improvement resulting from the second contribution. The effect of improved computation complexity only exists if prediction ratio < masking ratio. Lastly, according to Table 3 d), with the right number of fused feature maps, inter-block attention CAN improve the model, but the authors choose 12 as their default number of fused feature maps, which doesn’t improve performance over just adding Cross-Attn.

Concretely, I think the following additions could improve the paper:
- Introduce a comprehensive analysis of the individual contributions’ impact on performance, and also computational complexity if you want to highlight that, in a similar manner as proposed above
- Train both models for 1600 epochs and evaluate if the performance increase can be sustained

I’m willing to increase my score if my concerns are adequately addressed, and/or if the other reviewers list further convincing arguments for accepting the paper.

**Limitations:**

The authors have sufficiently addressed the limitations of their approach.

---

> ### Author Rebuttal · Authors · 2024-08-07
>
> We want to thank the reviewer for the detailed review. We provide responses via the discussion below.
>
> The most critical concern that the reviewer had was outlined in **Weakness 2**:
> > [W2] I don’t understand how the prediction ratio provides any benefit for better downstream performance.
>
> We apologize for the confusion regarding why we introduced a varying prediction ratio (partial reconstruction). Partial reconstruction does not intend to improve the performance in terms of downstream accuracy. Instead, it **allows our user to control and significantly reduce the computation required during training in terms of runtime and GPU memory usage with minimal impact on downstream performance**, as explained in L196-197. **Additionally, in the general rebuttal above, we present a derivation of the computational complexity as well as runtime statistics.**
>
> In addition to the absolute downstream performance that the reviewer focused on throughout the review, one of the ways to improve model performance is making training more **efficient** (i.e. reaching a similar downstream classification accuracy and segmentation accuracy as baselines with less training time and compute).
>
> Practically, the outlined improvement in efficiency not only **allows longer and more exploratory experiments to be run for improved downstream accuracy within the same compute budget** but also **makes pre-training more feasible in settings where compute resources are more constrained** (Table 10). We hope that the findings we had in the paper along with the improved pre-training efficiency can lead to more future research in improving visual pre-training.
>
> Given this, we address each of the questions individually:
> > [W1/Q1] “missing a more structure ablation of the individual contributions”
>
> We refactored Table 3, the individual contribution’s impact on performance. All time trials are averaged over 10 epochs with FlashAttention 2 enabled. Note that the runtime is benchmarked with 2x NVIDIA A100 GPUs while Table 10 uses 4x A100 (L 584), so the runtime roughly doubles compared to Table 10.
>
> | Method                         | Accuracy | Runtime (mins per epoch) |
> |-------------------------------|----------|---------------------|
> | MAE                           | 83.0     | 9.35              |
> | +Cross Attention              | 82.9     | 7.38              |
> | +Inter-block Attention*         | **83.3**     | 8.41              |
> | +Partial Reconstruction       | 83.2     | **6.32**              |
>
> \* Since in cross-attention, the keys and queries can take different values. This enables each decoder block to use different encoder features through inter-block attention. This would otherwise not be possible with self-attention decoder blocks in vanilla MAE.
>
> We will incorporate this table to improve the clarity of our work’s contributions and advantages.
>
> > [W3/Q2] “Train both models for 1600 epochs”
>
> We provide an experiment at 1600 Epoch in the table below. Both MAE and CrossMAE are trained with the same hyperparameters and on the ViT-B architecture. **CrossMAE still performs comparably while being faster at pre-training.**
>
> | | MAE | CrossMAE (0.25) | CrossMAE (0.75) |
> | - | - | - | - |
> | ImageNet Acc | 83.6 | **83.7** | **83.7** |
> | Runtime (mins per epoch) | 9.35 | **6.32** | 8.41 |
>
> > [W4] Table 1 is missing the CrossMAE ViT-H with 75% masking ratio
>
> Due to resource constraints and limited time for the rebuttal, we are not able to finish training ViT-H at 75% masking ratio. However, we did show that **even at 25% masking ratio our ViT-H performance surpasses the performance of MAE** (86.3% vs 85.8%), which is a setting requiring **even less compute** compared to 75% masking ratio.
>
> > [W5,6] “Contribution 2 and 3 don’t seem to be as well motivated in the introduction in comparison to Contribution 1; better performance (contribution 3) is a result of the technical contributions”
>
> We have restructured the list of contributions as below:
> 1. (left unchanged) **We present a novel understanding of MAE.** Our findings show that MAE reconstructs coherent images from visible patches not through interactions between patches to be reconstructed in the decoder but by learning a global representation within the encoder. This is evidenced by the model’s ability to generate coherent images and maintain robust downstream performance without such interactions, indicating the encoder effectively captures and transmits global image information.
>
> 2. Given our discovery that the encoder in MAE already captures a comprehensive global representation, **we propose replacing self-attention layers with cross-attention** to aggregate the output tokens of the encoder into each input token within the decoder layers independently, thereby eliminating the need for token-to-token communication within the decoder.
>
> 3. Finally, **we leverage additional properties of cross-attention to achieve an even better performance-efficiency trade-off**. CrossMAE's ability to independently reconstruct masked tokens allows us to process only a fraction of the masked patches during training, significantly improving efficiency. Furthermore, the use of cross-attention enables different decoder blocks to utilize distinct encoder features, enhancing the performance-compute trade-off through inter-block attention. This approach achieves comparable or superior results in image classification and instance segmentation tasks across various model sizes (from ViT-S to ViT-H) while reducing computational demands compared to MAE.
>
> **In light of the clarifications and analyses we provide, we would like to ask if you might be open to increasing your assessment score, and if there are any additional concerns or questions that we can address for you!**

---

> ### Author Response · Authors · 2024-08-09
> **Additional experiment for CrossMAE ViT-H with 75% masking ratio**
>
> > [W4] Table 1 is missing the CrossMAE ViT-H with 75% masking ratio
>
> Our CrossMAE ViT-H run with 75% masking ratio just finished. The updated comparison for ViT-H is shown in the table below. **Our method, either with 25% or 75% masking ratio, surpasses MAE in terms of ImageNet fine-tuning performance on ViT-H, justifying the scalability of our method.** The result further justifies that reconstructing only 25% of the masked tokens brings large efficiency gains, as analyzed in the general response, but has a marginal impact on downstream performance.
>
> | Method          | ViT-S | ViT-B | ViT-L | ViT-H |
> |-----------------|-------|-------|-------|-------|
> | Supervised | 79.0  | 82.3  | 82.6  | 83.1  |
> | MAE        | 78.9  | 83.3  | **85.4**  | 85.8  |
> | CrossMAE (25%)  | 79.2  | 83.5  | **85.4**  | 86.3  |
> | CrossMAE (75%)  | **79.3**  | **83.7**  | **85.4**  | **86.4**     |

---

> > ### Author Response · Authors · 2024-08-12
> > **A gentle reminder - 2 days left for the author-reviewer discussion**
> >
> > Dear reviewer,
> >
> > We wanted to ask if you had a chance to check our response with additional details and clarifications, a more structured ablation study, a comparison of model performance at 1600 epochs, CrossMAE ViT-H with 75% masking ratio, and an updated list of contributions.
> >
> > Please also consider checking our general response, which includes a more detailed analysis of the improved efficiency of CrossMAE.
> >
> > Please let us know if we addressed your concerns and/or if any further information or experiments would be helpful. We would be happy to provide them.
> >
> > Many thanks!
> > Authors

---

> > > ### Comment · Reviewer_cyeA · 2024-08-12
> > > **Response to rebuttal**
> > >
> > > Dear authors,
> > > Thanks for addressing my concerns regarding your paper.
> > > For my final rating, I will take into consideration the arguments which arise during the upcoming AC-Reviewers discussion phase. Overall, I lean towards increasing my score.
> > >
> > >  Best regards!

---

> > > > ### Author Response · Authors · 2024-08-12
> > > >
> > > > Thank you for your tendency to increase your score! Please feel free to let us know if any other question arises before the rebuttal deadline, and we would be happy to answer them.

---

### Official Review · Reviewer_gKk3 · 2024-07-12

**Soundness:** 3
**Presentation:** 3
**Contribution:** 3
**Rating:** 7
**Confidence:** 4

**Summary:**

This paper presents CrossMAE, a methodology for improving pre-training efficiency over that of MAE for an encoder. The paper motivates its approach by presenting visual evidence that, in standard MAE pre-training, masked tokens attend to other masked tokens significantly less than to non-masked (aka, visible) tokens. Using this motivation, the paper then presents CrossMAE, which differs from MAE largely in that it replaces the MAE self-attention with cross-attention between the masked tokens and a learnable weighted combination of the encoder feature maps. This aspect decouples queries from keys and values (which is not the case in MAE), which the paper then exploits to allow only some (but not necessarily all) mask tokens to be used during reconstruction to pre-train the model. The paper presents an analysis of which encoder block features are optimal to cross attend with each decoder block, and it presents ablation studies on multiple design decisions. Finally, it presents visual and fine-tuning results showing comparable performance to MAE and similar methods.

**Strengths:**

This paper motivates CrossMAE well by showing evidence of a potential inefficiency in MAE (self-attention) and then presenting an approach to remedy it (cross attention). I particularly like how the paper delves even deeper, though: instead of stopping at the level of replacing self-attention with cross-attention, it then points out that this choice allows for a significantly fewer number of masked patches to have to be reconstructed, which reduces flop count significantly. The ablations in Table 3 are fairly thorough and answered some questions I have developed. The performance of CrossMAE appears comparable to other SOTA methods but with significantly more efficient pretraining.

**Weaknesses:**

1) In Fig 1b, IIUC, for one particular mask token, the two $\mu$'s are the respective attention values averaged over all transformer blocks and all masked/non-masked tokens. If this is the case, my concern is that by averaging over all transformer blocks, variations in the attention is being hidden. Naively, I would think that for early blocks, the attention due to masked tokens would be small (as the paper concludes) but becomes larger for the later blocks (since now the masked tokens have actual useful signal in them). Did you consider this?

2) I do not follow why CrossMAE does not need an MLP at the end to convert final decoder tokens back to raw pixels. Line 218 says that the inputs to the first encoder block are included in the feature maps for cross attentions. Does this cause a final MLP to not be used?

3) Less critical:
  3a) Fig 1b should point the reader to Section A.1. I spent much of my reading confused about what $\mu$ is.
  3b) Fig 4a should have a different number of decoder layers than encoder layers. When I saw this figure, I immediately wondered why a decoder block wasn't being paired with feature maps from its "partner" encoder. I had to wait until lines 204-207 to get an explanation of why this doesn't work.
  3c) Line 187 references a "second question" in Sec 3.1, which doesn't exist as far as I can tell.
  3d) Fig 4a shows the "vanilla" version of Cross MAE, where the final encoder layer feature maps are attended with all decoder layers. But the paper presents results exclusively (?) on the version that uses a learned combination of the feature maps. Anyway, the figure confused me. Maybe I just didn't understand what the solid arrows vs dotted ones are supposed to represent.

**Questions:**

See "Weaknesses".

**Limitations:**

No weaknesses are specifically addressed. But as this paper is essentially an optimization to MAE, I'm not sure this question is relevent.

---

> ### Author Rebuttal · Authors · 2024-08-07
>
> Thank you for your valuable questions and suggestions! We provide responses via the discussion below:
>
> > [W1] “By averaging over all transformer blocks, variations in the attention may be hidden. Naively, (the reviewer) would think that for early blocks, the attention due to masked tokens would be small but becomes larger for the later blocks.”
>
> This is indeed an interesting hypothesis! We test this hypothesis on the pre-trained MAE and report the cross-attention and self-attention magnitude at each individual block in the table below.
>
> | Block | Attention to visible tokens | Attention to masked tokens |
> |-------|---------------------------|--------------------------|
> | 1 (closest to the encoder)    | 0.196                     | 0.042                    |
> | 2     | 0.149                     | 0.058                    |
> | 3     | 0.181                     | 0.047                    |
> | 4     | 0.153                     | 0.057                    |
> | 5     | 0.193                     | 0.043                    |
> | 6     | 0.161                     | 0.054                    |
> | 7     | 0.207                     | 0.038                    |
> | 8 (closest to the reconstruction)    | 0.179                     | 0.048                    |
> | Sum (reported in the paper) | 1.418                    | 0.388                    |
>
> Based on these results, we do not observe a significant increase in the magnitude of the attention to masked tokens for the later decoder blocks. In addition, our observed pattern that “the magnitude of attention is larger in masked tokens’ cross-attention to visible tokens than in masked tokens’ self-attention” is shared across all decoder layers.
>
> > [W2] “I do not follow why CrossMAE does not need an MLP at the end to convert final decoder tokens back to raw pixels.”
> > Line 218 says that the inputs to the first encoder block are included in the feature maps for cross attentions. Does this cause a final MLP to not be used?
>
> Sorry for the confusion! We **do** need a final MLP to convert the final **decoder** token back to raw pixels, which is consistent with MAE.
>
> Line 218 refers to the process of passing the aggregated feature of each **encoder** block to each decoder block rather than converting the final decoder features to raw pixel values, where we treat the input feature after patchfication and linear projection as one of the encoder features to be aggregated. We will update the paper to clarify this further.
>
> > [W3] 3.a-d
>
> Thank you so much for pointing these out! We have made the following changes to the paper to provide clarifications.
>
> > For 3.a: “Fig 1b should point the reader to Section A.1. I spent much of my reading confused about what μ is.”
> We updated the third paragraph to clarify what μ is and updated the figure captions. Now the third paragraph and the caption reads as below (with main changes in **bold**):
>
> Line 28-33:
> We decompose the decoding process of each masked token into two parallel components: self-attention with other masked tokens, as well as cross-attention to the encoded visible tokens. If MAE relies on self-attention with other masked tokens, its average should be on par with the cross-attention. Yet, the quantitative comparison in Figure 1.(b) shows **the average magnitude of masked token-to-visible token cross-attention** (μ=1.42) in the MAE decoder evaluated over the entire ImageNet validation set far exceeds **that of masked token-to-masked token self-attention** (μ=0.39). **We describe the attention calculation in Section A.1.**
>
> Figure 1 Caption:
> (B) MAE reconstructs a masked token (marked by a blue star) by attending to both masked tokens (B.Left) and visible tokens (B.Right). A quantitative analysis of the ImageNet validation set reveals that masked tokens in MAE attend disproportionately to visible tokens compared to other masked tokens (**average attention magnitude** μ=1.42 vs μ=0.39, respectively). This observation raises questions about the necessity of attention among masked tokens themselves.
>
> > For 3.b/d: “Fig 4a: should have a different number of decoder layers than encoder layers … Fig 4a shows the "vanilla" version of CrossMAE, but the paper presents results exclusively on the version that uses a learned combination of the feature maps.”
>
> Thanks for the suggestions. We have updated Figure 4a accordingly and attached the updated figure in the PDF in the general rebuttal (see the figure for Reviewer gKk3).
>
> > For 3.c: “Line 187 references a "second question" in Sec 3.1, which doesn't exist as far as I can tell.”
> Now Line 187 reads: “Rather than decoding the reconstruction for all masked locations, we only compute the reconstruction on a random subset of the locations and apply the loss to the decoded locations.”
>
> **Thank you once again for your positive feedback! If you have any further questions, please don’t hesitate to ask!**

---

> > ### Comment · Reviewer_gKk3 · 2024-08-12
> >
> > Thank you for your detailed responses to my concerns. From W1, I conclude that my intuition is far from always correct. I appreciate the study you made.

---

### Author Rebuttal · Authors · 2024-08-07

We would like to thank all the reviewers for their thoughtful reviews as well as encouraging feedback. We are especially glad that the reviewers believe that our ablations are **“fairly thorough”** (Reviewer gKk3), the paper is **“well motivated through a practical observation”** and **“well written and technically sound”** (Reviewer cyeA), and our work offers **“main messages that are solid and insightful”** (Reviewer yYWM). We respond to some common concerns in the general response below. We are more than happy to discuss with the reviewers to address any additional questions during the discussion period.

## The goal of decoupling the prediction ratio from the masked ratio (partial reconstruction) [Reviewer cyeA, yYWM]
Partial reconstruction does not intend to improve the performance in terms of downstream accuracy. Instead, it **allows our user to control and significantly reduce the computation required during training in terms of runtime and GPU memory usage with minimal impact on downstream performance**, as explained in L196-197. We refer the reviewers to the response below for a detailed analysis of computational complexity.

Concretely, we compare the ImageNet classification accuracy, runtime, and memory requirements in the table below (the setting follows Table 1 and the runtime is measured on 2x A100 80GB with Flash-Attention 2 [1] enabled for all three models; gradient accumulation is set to 2):

| Method (prediction ratio) | MAE (0.75) | CrossMAE (0.75) | CrossMAE (0.25) |
|-----------------------------|------------|-----------------|-----------------|
| Accuracy                | 83.0      | **83.3**           | 83.2           |
| Runtime in mins per epoch | 9.35       | 8.41            | **6.32**            |
| Memory (MB per GPU) | 68386 | 57987 | **36805** |

Furthermore, **we would like to underscore that partial reconstruction depends on our proposed use of cross-attention decoder blocks and is not directly applicable to vanilla MAE for improved efficiency.**
In vanilla MAE, each masked token attends to other masked tokens in the decoder blocks. Consequently, removing any masked tokens will change the decoded values of the remaining ones. However, CrossMAE generates each masked token based solely on the visible tokens, without considering other masked tokens. This means the reconstruction of a masked patch would not be affected based on which subset of masked tokens are decoded. As a result, the loss applied to this subset serves as an unbiased estimate of the original loss that would be applied to all reconstructed tokens. This approach minimizes performance impact while significantly reducing computational requirements and runtime.

## Analysis on computational complexity and effectiveness of small prediction ratio [Reviewer cyeA, yYWM]
Since vanilla MAE uses a self-attention decoder, the computation complexity of the decoder is quadratic with respect to the total number of tokens. Using cross-attention reduces the computation complexity to be linear with respect to the number of visible tokens and the number of masked tokens. By varying the number of masked tokens through partial reconstruction, the complexity can be further reduced by only decoding a subset of the masked tokens.

Formally, let the total number of visible tokens be $N$, latent dimension be $d$, with a mask ratio $p$ and prediction ratio $\gamma$. The complexity of attention in the MAE decoder is on the order of $N^2d$. In CrossMAE, the visible tokens (of length $(1-p)N$) serve as the queries, and the masked tokens (of length $\gamma N$) serve as the keys and values. The resulting attention complexity in CrossMAE decoder is $(1-p)\gamma N^2d$. In Table 1, the CrossMAE variant that decodes all masked tokens, or CrossMAE (0.75) uses $p=0.75$ and $\gamma=0.75$ (i.e., full prediction with $p=\gamma$), which gives $\frac{3}{16}N^2d$. On the other hand, CrossMAE (0.25) uses $p=0.75$ and $\gamma=0.25$, which gives $\frac{1}{16}N^2d$, further reducing the complexity.

The reduced computational complexity does translate to **faster training and lower memory requirements**, as shown in the table above, and is orthogonal to other acceleration techniques such as Flash-Attention 2 [1].

[1] Tri Dao. FlashAttention-2: Faster attention with better parallelism and work partitioning. 2023. 2, 3

---

### Decision · Program_Chairs · 2024-09-25

**Decision:**

Reject

**Comment:**

After the rebuttal and discussion period, the submission received mixed reviews. The AC carefully reviewed the paper and agrees with the concerns raised by Reviewer yYWM, particularly regarding the lack of experimental comparisons with state-of-the-art MAE approaches, especially those that also consider inter-patch relations during self-supervised learning. Simply comparing the model's performance with the original MAE is insufficient, as the presented results did not demonstrate a significant performance improvement.

Given this concern, the AC could not support acceptance. The authors are however strongly encouraged to revise the submission based on the feedback for consideration at a future venue.